# Systematic Review of Dementia Support Programs with Multicultural and Multilingual Populations

**DOI:** 10.3390/geriatrics7010008

**Published:** 2021-12-30

**Authors:** Abriella Demanes, Katherine T. Ward, Amy Tu Wang, Mailee Hess

**Affiliations:** 1Department of Medicine, Harbor-UCLA Medical Center, Torrance, CA 90502, USA; KWard@dhs.lacounty.gov (K.T.W.); AWang7@dhs.lacounty.gov (A.T.W.); MHess@dhs.lacounty.gov (M.H.); 2Section of Geriatrics, Division of General Internal Medicine, Department of Medicine, Harbor-UCLA Medical Center, Torrance, CA 90502, USA

**Keywords:** dementia, care coordinator, minority

## Abstract

Background: Dementia care programs have become more common due to a growing number of persons living with dementia and lack of substantial benefit from pharmacologic therapies. Cultural and language differences may present barriers to access and efficacy of these programs. In this article, we aimed to systematically review the current literature regarding outcomes of dementia care programs that included multicultural and non-English speaking populations. Methods: A systematic review was conducted using four scientific search engines. All studies included in the review are English language, randomized control trials evaluating various care coordination models. The initial search strategy focusing on studies specifically targeting multicultural and non-English speaking populations resulted in too few articles. We expanded our search to articles that included these populations although these populations may not have been the focus of the study. Results: Seven articles met inclusion criteria for final review. Measured outcomes included emergency room use, hospitalizations, provider visits, quality of life indicators, depression scores, and caregiver burden. Conclusions: Dementia care programs demonstrate significant ability to provide support and improve outcomes for those living with dementia and their caregivers. There is limited research in this field and thus opportunity for further study in underserved and safety net populations including more high-quality randomized controlled trials with larger sample sizes.

## 1. Introduction

According to the World Alzheimer Report, it is estimated that by 2030 the global number of persons living with dementia will be 74.7 million and increase to 131.5 million by 2050 [1]. Those living with dementia may suffer devastating outcomes including inappropriate and potentially harmful medication use, frequent hospitalizations, and aggressive end-of-life care inconsistent with their goals of care [2,3,4,5,6]. Dementia has become a public health issue that affects not only those with dementia but also those who love and care for them. Caregivers experience high levels of stress and burden, which can negatively impact their physical and emotional health [4,5,7,8].

Navigating the healthcare system for persons with dementia is a challenge and usually falls on the caregiver. To mitigate these challenges, dementia care programs and dementia care-coordinators are increasingly used to address the interdisciplinary needs of those with dementia and their caregivers [6,9]. Studies show that caregivers who can access support and resources experience benefits, including improved understanding of dementia, care plans, and reduced caregiver depression, fatigue, and feelings of isolation [8,10,11]. Whereas using dementia medications has not led to substantial improvements in clinical meaningful outcomes, dementia care programs have shown benefit [12]. However, the structure, components, and efficacy of these programs vary. Furthermore, barriers to access to these programs exist, including language, culture, and geographic disparities [6,7,8]. To better understand the components and the effectiveness of these types of programs, we conducted a systematic review and evaluation of the current published literature pertaining to dementia care programs that included multicultural and multilingual populations and their outcomes.

## 2. Methods

### 2.1. Data Sources

A systematic review was conducted to investigate outcomes of dementia support programs for persons with dementia and their caregivers. While our initial search was for studies that targeted multicultural or non-English speaking populations, there were no articles that specifically met this criterion. Therefore, we changed our search to articles that included multicultural and non-English speaking populations. Literature search was conducted using PubMed, MEDLINE, CINAHL, and PsycInfo. Key terms included dementia and care coordinator. For PubMed, the “similar articles” feature was used to expand the search. Searches were then limited to peer reviewed journal articles, which were written in English and published after 2005.

### 2.2. Inclusion and Exclusion Criteria

Articles were included if the article was a randomized control trial and investigated an intervention that targeted support for persons with dementia and/or their caregivers. The trial also had to include multicultural or multilingual populations. Observational studies, reviews, editorials, commentaries, and case studies were excluded. Articles published outside the United States were included.

### 2.3. Study Selection

The primary author reviewed the titles and abstracts of all retrieved articles to assess for relevance prior to reviewing the article in full. Articles were included for full review if it was unclear from their title or abstract if a specific intervention was used in an interventional design. During full review, articles were eliminated if they investigated the same dementia support program. Only the article that scored the highest quality based on the Modified Downs and Black checklist was included. A total of seven articles were included in the final review (Figure 1). Due to the wide range of interventions and outcomes examined it was unfeasible to pool results for a quantitative meta-analysis.

### 2.4. Data Abstraction

Data abstraction was performed on seven articles by the primary author and included: population, clinical setting, sample size, intervention and comparison group, measured outcome, and major findings (Table 1). The number of multicultural or non-English speaking populations included in the study was also noted.

### 2.5. Quality Appraisal

The seven studies included in this review were systematically appraised using the modified Downs and Black checklist. This tool may be used to evaluate both randomized and non-randomized control trials by scoring quality of reporting, external validity, bias, confounding variables, and power, although this review article only includes randomized controlled trials [13]. The maximum score for this checklist is 28. Modified Downs and Black score ranges mirrored those reported in previous studies: ≥20 very good; 15–19 good; 11–14 fair; ≤10 poor [14].

## 3. Results

### 3.1. Article Selection

Utilizing the search strategy described above, a total of 1404 articles were initially identified. Articles were excluded if they were published prior to 2005 or not written in English. After applying these filters, 1193 articles remained. After removing articles that were not peer reviewed and not randomized controlled trials, 67 articles remained. The titles and abstracts of these 67 articles were screened and 31 articles were retained for full screening. After removing duplicates and reviewing the articles for relevance and inclusion of multicultural or non-English speaking populations, seven articles remained for full review.

### 3.2. Type of Studies

#### 3.2.1. Study Design

The final review included studies that were published between 2006 and 2019. Study durations ranged from one month [8] to two years [6]. Although all studies were randomized, one study did not provide details surrounding randomization [8], one study was randomized at the level of the provider [15], another study was randomized at the level of the study site [16], while all other studies were randomized at the level of the participant [6,17,18,19]. While most studies were conducted in the US, two were conducted outside of the US including Australia [17] and Mexico [8].

#### 3.2.2. Setting

All interventions occurred at the person’s home except for in the study conducted by Callahan et al., which included a mixture of home-based support and office visits. Support interventions included care coordination, needs assessments, linkage to resources, providing education, emotional support, or a combination of these. Additionally, these interventions were provided in varied ways including use of a culturally sensitive educational website [8], use of a therapist or certified interventionalist to teach problem solving techniques [19], or more commonly, use of a care manager to provide care coordination with needs assessments, screenings, education, and linkage to resources [6,15,16,17,18].

#### 3.2.3. Study Population

Multicultural or multilingual participants were included in all studies. Two of the seven studies specifically targeted multicultural populations. These two studies evaluated different forms of technology to support persons with dementia. In the Czaja et al. study which evaluated a videophone technology, 50% of their study population identified as Hispanic/Latino with the other half of their population identifying as African Americans. The second study, Pagan Ortiz et al., evaluated a web-based platform in a Hispanic population. Callahan et al. had almost 50% of their study population identifying as Black/African American although multicultural populations were not specifically targeted in this study. Possin et al. included some multicultural groups, but their numbers tended to be small with only 4% identifying as African American, 10% Hispanic/Latino, and 10% Cantonese in their investigation of a phone-based support program.

Five studies reported non-English speaking dyad members. 86% of the Xiao et al. dyad population (language not specified), 7.5% in Possin et al. (2% Cantonese, 5.5% Spanish), and 100% of Pagan-Ortiz et al. (Spanish). Czaja et al. included non-English speaking populations but did not report the number of individuals who spoke Spanish.

#### 3.2.4. Outcomes

Study outcomes fell into two groups: health care utilization or clinical outcomes. Health care utilization was evaluated in four studies, specifically emergency room use, hospitalizations, or provider visits [6,15,16,18]. Clinical outcomes, including quality of life indicators, depression scores, and caregiver burden scores, were evaluated in five studies [6,8,15,17,19].

## 4. Identification of Key Themes

### 4.1. Care Team Members

Three of the five studies used a care manager to provide care coordination used licensed clinical persons in the role of the care manager, such as social workers, registered nurses, and nurse practitioners. Only two studies used nonclinical persons as the care manager. In one study, the nonclinical person was supported by an interdisciplinary clinical team consisting of an RN and geriatric psychiatrist [18]. In the other, Possin et al. utilized an unlicensed care team manager who was provided with 40 h of training and had access to higher level clinical providers (e.g., RNs, pharmacists, social workers) if needed. While care coordination had mixed results with improvements in health care utilization, care coordination demonstrated positive clinical outcomes regardless of whether the care coordinator was a licensed clinical person.

### 4.2. Health Care Utilization

Five studies investigated care recipient’s health care utilization, which included number of ED visits, utilization of acute care, inpatient, outpatient, and home-and community-based services. Bass et al. found a significant decrease in both ED visits and hospitalizations in the intervention groups while Possin et al. showed a significant decrease in ED visits, but not hospitalizations. Amjad et al. showed no significant difference between number of inpatient or outpatient services. Callahan et al. did not show improvements in nursing home placement.

### 4.3. Clinical Outcomes

Studies evaluating clinical outcomes showed consistently positive results. Two studies resulted in less caregiver depression after participation in a dementia care program [6,8]. Three studies also found a reduction in caregiver burden [6,8,19]. Callahan et al. showed significant improvement in behavioral neuropsychiatric inventory (NPI) scores and caregiver stress and Xiao et al. reported improved quality of life measures. Of note, Czaja et al. found that almost three times as many participants in the intervention group reported significant improvements in positive aspects of caregiving after participating in an at-home, technology-based education platform for dementia care. These studies varied in their intervention from face-to-face visits [17] to virtual methods, including telephone-based [6,17], video-based [19], and web-based [6,8] visits.

### 4.4. Type of Educational Materials

All studies involved providing caregivers with education on how best to care for their family or loved one living with dementia. However, the type of education differed. Possin et al. educated caregivers about dementia [6] whereas four studies focused on strategies for managing challenging behaviors exhibited by persons with dementia [15,17,18,19]. Pagan-Ortiz et al. provided both types of education.

### 4.5. Multicultural or Non-English Speaking Participants

Although all studies included populations known to have barriers in accessing care including racial minorities or non-English speaking participants, only two studies investigated outcomes specific to these populations [8,19].

### 4.6. Quality of Studies

The Black and Downs scores of the seven studies included in this review ranged from 10 to 23 with a median score of 19. Based on this method of appraisal four studies received a “very good” quality rating [6,15,17,18], two studies were “good” [16,19], none were “fair”, and one study was “poor” [8].

## 5. Discussion

With the increasing population of older adults living in the U.S and corresponding rise in numbers of adults living with dementia, there has been a growing interest in and need for care interventions for persons with dementia (PWD) and their caregivers. Collaborative care models and multicomponent interventions have been shown to improve caregiver burden and depression, PWD quality of life, and decreases in resource utilization, such as ED visits, hospitalizations, and nursing home placement. Currently there is little information to determine whether these interventions are equally effective for multicultural populations and rural communities/communities with low resources. This review originally intended to review the efficacy of these interventions in multicultural or non-English speaking populations. However, this strategy was too limiting, and we expanded this review to evaluate studies that included these populations.

This systematic review highlights the value of dementia care programs in a variety of domains ranging from psychosocial and quality of life measures of persons with dementia and caregivers alike [6,8,17,19] to health care utilization [6,16]. A limitation identified in this review is that although most studies included for review are randomized controlled trials, not all studies were randomized at the level of the participant.

It is well established that caring for persons with dementia results in physical and psychological strain on caregivers. Challenges include helping with activities of daily living, managing psychological and behavioral symptoms of those with dementia, and perceived changes in relationship between caregivers and the person with dementia [9,10,17]. As the disease severity progresses over time, caregivers require ongoing assistance to help address challenges regarding education, daily care practices, other care services, as well as their own emotional and psychological well-being [8,11,17]. These needs may be addressed by dementia care programs.

In addition to the stressors of caring for a loved one with dementia, underrepresented multicultural populations and non-English speaking caregivers face added barriers to care. Communication barriers have been identified as a barrier to non-English speaking caregivers and families from seeking supportive services [14]. Furthermore, many resources for caregivers are designed to target the predominant culture and those who speak English [17]. Mixed race populations are understudied in trials regarding dementia care programs [15,20]. In conducting this review, it was apparent that there is limited research targeting underserved and safety net populations in this area. Although a number of studies mentioned minority populations, only two specifically targeted minority populations [8,19], further highlighting the need for more research in this area.

In the two studies that included these populations, Pagan-Ortiz et al. utilized a website to provide culturally sensitive dementia education and support for Hispanic families, and Ceja et al. used certified interventionalists to teach problem solving strategies to Hispanic and African American caregivers. The Pagan-Ortiz et al. study showed no statistically significant outcomes in self-mastery, social support, caregiver burden, or depression. Of note, study participants in Pagan-Ortiz et al. were mostly located in Mexico or Puerto Rico. Only five participants were recruited in Massachusetts. Hispanic populations in the United States face different barriers to care than in Mexico or Puerto Rico where Hispanic culture is predominant. Of the interventions listed in Table 1, use of a culturally sensitive website is the least intensive and requires more initiative on the part of the caregiver to engage with the program. Ceja et al. showed decrease caregiver burden and increased appreciation for the positive aspects of caregiving and satisfaction with social support. This more intensive intervention showed positive outcomes in underrepresented populations. This is the only article we found that showed positive outcomes specific to multicultural populations. This highlights a need for further randomized trials in populations that face barriers to accessing care in the U.S. The other studies that included multicultural or non-English speaking populations did not report outcomes specific to these specialized populations as the sample size for these populations was not large enough.

Another study, carried out by Chodosh et al., was not included as it was not a randomized control trial, but offered support for low-income Hispanic and Black communities in Los Angeles that partnered with the Alzheimer’s Association and conducted either in person or phone visits for care coordination. This study showed improved caregiver burden and problem behaviors [21], which is promising, but again highlights the need for further randomized controlled trials of dementia support programs in these populations.

Despite the lack of diversity in the trials presented here, several dementia care programs, whether through face-to-face clinical coordinators or by virtual means, have shown substantial benefit in quality of life measures. In one study conducted by Callahan et al., participants received 1 year of care management by an interdisciplinary team led by an advanced nurse practitioner integrated within the primary care setting. This study demonstrated that a comprehensive care approach resulted in clinically significant improvements in behavioral and psychological symptoms of dementia and reduction in caregiver stress. Ensuring that caregivers are properly supported has demonstrated positive outcomes for both persons with dementia and their caregivers [6].

Care coordinators may also assist in the fragmentation of medical care, provide resources, and potentially reduce healthcare costs [18]. In general, those living with dementia have higher rates of ED visits and hospitalizations, which may yield undesired consequences, including delirium, falls, medical complications, functional decline, and nursing home placement [16,22,23,24,25,26,27,28]. In a study by Bass et al., a program called Partners in Dementia, which was a collaboration between Veterans Affairs medical centers and the Alzheimer’s Association created to address the needs of persons with dementia and their caregivers, showed a reduction in hospital admissions and ED visits with corresponding healthcare costs [6,16].

As the number of individuals with dementia increases and caregivers are recognized as a precious resource, using all available tools to assist caregivers may mitigate the challenges they face [29,30]. Technology allows for the opportunity to provide tailored support and evidence-based interventions to caregivers [29,31]. Online communities are a feasible way for geographically dispersed groups to meet online for education, support, and social connection [8]. Possin et al. created a telephone based collaborative dementia care program called Ecosystem to provide education, support and care coordination to caregivers and persons with dementia. This study found that dementia care management delivered over the telephone and internet may reduce growing societal and economic burdens of dementia. In a study by Xiao et al., coaching and support provided to caregivers over the phone improved caregivers’ sense of competence in managing dementia and their mental well-being. Additionally, programs that utilize technology allow for the opportunity to reach individuals in rural areas that may otherwise not have the opportunity to participate in a dementia care program [29]. Technology is more easily accessible for persons with dementia and caregivers who otherwise would be unable to access these resources due to barriers in transportation and proximity to physical resources.

## 6. Conclusions

In summary, dementia care programs provide significant benefit to those living with dementia and their caregivers. Dementia care programs, whether through face-to-face coordination or virtual means, show significant promise in providing improvements in access to resources and quality of life measures for persons with dementia and caregivers alike. Furthermore, virtual based programs may be particularly helpful in underserved or safety net populations as this may improve access to dementia care programs. As research in this field is limited, more high quality studies using larger sample sizes are needed. Additionally, there is a particular need for further research in the development and efficacy of dementia care programs in multicultural and multilingual populations.

## Figures and Tables

**Figure 1 geriatrics-07-00008-f001:**
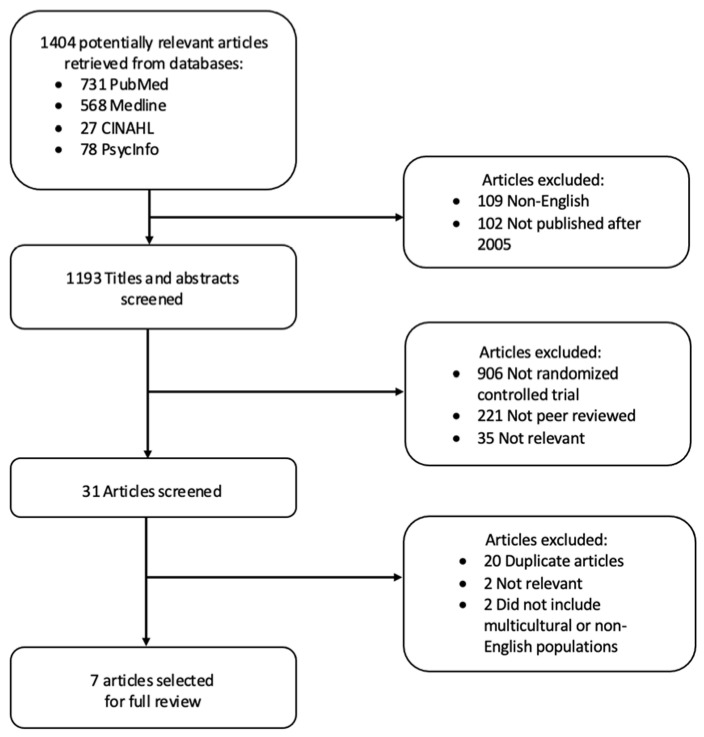
Results of research strategy.

**Table 1 geriatrics-07-00008-t001:** Characteristics of included studies.

Article	Population	Clinical Setting	Sample Size	Intervention and Comparison Group	Measured Outcomes	Major Findings	Black and Downs Score
Bass et al., 2015	Community-dwelling, veterans ≥ 60 yo with dementia and their caregivers located in 5 major US cities (Boston, MA, Houston, TX, Providence, RI, Oklahoma City, OK, Beaumont, TX) * multicultural populations included	Virtual (e.g., telephone, mail and email)	*n* = 508 (total) *n* = 299 (intervention) *n* = 187 (control) * 19% identified as member of multicultural group, however *n* not specified	Bachelor or Masters level SW or RN Veterans Affairs (VA) coordinator and Alzheimer’s Association (AA) coordinator collaborated to provide guidance for veterans and caregivers using standardized protocols via at least monthly phone calls. VA coordinator addressed medical-related concerns. AA coordinator addressed caregiver’s nonmedical concerns. Comparison group: no assistance from care-coordination program. Received educational materials.	Number of veterans’ hospital admissions and emergency department (ED) visits in persons with dementia over 12 months.	Veterans with dementia who received assistance from care coordinators had fewer hospital admissions and ED visits than comparison-group veterans. There were no differences in the likelihood of hospital admission or ED use.	16
Xiao et al., 2016	Caregivers ≥ 18 yo from minority groups who cared for a community-dwelling person with dementia (PWD) from the same multicultural group located in Metropolitan Adelaide, South Australia. * Non-English speaking participants included	Home visits and virtual (e.g., telephone)	*n* = 61 (total) *n* = 31 (intervention) *n* = 30 (control) * *n* = 53 (non-English speaking)	Care-coordinator with varied backgrounds (RN, SW, Community Home Care Certificate holders) who have cultural and linguistic concordance with caregivers provided support by screening for caregiver needs through home visits and phone calls, referring caregivers to services, and education programs. Caregivers also kept diary of unmet needs. Comparison group: no assistance from care coordinator.	Questionnaires addressing caregiver’s competence, quality of life (physical vs. mental), dependence level of care recipients, and satisfaction with care support.	The intervention group showed a significant increase in the caregivers’ sense of competence and mental components of quality of life. There were no significant differences in the caregivers’ physical components of quality of life.	20
Amjad et al., 2018	Community-dwelling adults ≥ 70 yo with cognitive impairment residing in North West Baltimore * multicultural populations included	In-home visits (at baseline and 18 months) and at least one monthly contact (e.g., telephone or in-person)	*n* = 303 (total) *n* = 110 (intervention) *n* = 193 (control) * *n* = 87 (Black/African American or other race)	18-month care coordination intervention provided by community-based, nonclinical care coordinators that were supported by interdisciplinary clinical team. Care coordination with nonclinical memory care coordinator + RN + geriatrician + psychiatrist (no PMD involvement) who provide education, skill building, linkage to services, informal counseling and care monitoring for 18 months. Comparison group: no assistance from care-coordinator.	In-person, self reported interviews administered at baseline, 9 months, and 18 months to assess utilization of acute care/inpatient, outpatient, and home-and community-based services.	No significant group differences in acute care/inpatient or total outpatient services use. Intervention group had significantly increased outpatient dementia/mental health visits from 9 to 18 months compared to controls. Intervention group had more home and community-based support service use from baseline to 18 months.	20
Possin et al., 2019	Community-dwelling persons with dementia-caregiver dyads ≥ 45 yo located in 3 US states (California, Iowa, and Nebraska). * Non-English speaking, multicultural, and rural participants included	Virtual (e.g., telephone and internet-based supportive care)	*n* = 780 (total PWD and caregiver dyads) *n* = 512 (intervention) *n* = 268 (control) PWD: * *n* = 31 (Spanish) * *n* = 16 (Cantonese) * *n* = 82 (Hispanic or Latino) * *n* = 31 (African American) Caregivers: * *n* = 43 (Spanish) * *n* = 16 (Cantonese) * *n* = 83 (Hispanic or Latino) * *n* = 32 (African American) * *n* not included for rural populations	Unlicensed care team navigator with 40 h of training provided telephone-based screening, support, education, and care coordination. Nurse, social worker and pharmacist provided support to care team navigator. Comparison group: no assistance from team navigator. They were offered contact info for Family Caregiver Alliance, Alzheimer’s association and area agencies on aging.	Primary outcome measure: Quality of Life in Alzheimer’s Disease based on caregiver survey of person with dementia. Secondary outcomes: frequencies of PWDs’ use of emergency department, hospitalization, ambulance services, caregiver depression, and caregiver burden.	Compared with usual care, intervention group showed improved quality of life for persons with dementia, reduced emergency department visits, and reduced caregiver depression and caregiver burden.	21
Pagán-Ortiz et al., 2014	Community-based Hispanic caregivers of PWD located in Puerto Rico, Mexico, or Massachusetts. * Non-English and multicultural populations included	Virtual (e.g., web-based)	*n* = 72 (total) *n* = 15 (intervention group that completed both pre-and post-test) *n* = 17 (control group that completed both pre-and post-test) * *n* = 72 (Spanish) * *n* = 72 (Hispanic)	Intervention group participated in 4 group sessions devoted to teaching them about features of a website that provides online education and support for Hispanic families, and professional caregivers of people with dementia. Comparison group: participated in 2 group sessions where they received printed Spanish educational materials on Alzheimer’s caregiving.	Caregivers were surveyed using pre- and post-test and assessed for sense of self-mastery, social support, burden, and depression symptomatology.	No outcomes were statistically significant	10
Callahan et al., 2006	Community-dwelling adults from two primary care practices in Indianapolis that met diagnostic criteria for Alzheimer’s disease and their caregiver. * multicultural population included	Primary care clinic, virtual (e.g., telephone-based)	*n* = 153 (total) *n* = 84 (intervention) *n* = 69 (control) * *n* = 75 (Black/African American)	Intervention group received 1 year of care management by an interdisciplinary team led by an advanced NP integrated within primary care setting who provided education on communication skills, caregiver coping skills, legal and financial advice, and implementation of behavior protocols when behaviors became an issue. Comparison group: augmented usual care without assistance from interdisciplinary team.	Neuropsychiatric Inventory (NPI) measured at baseline and at 6, 12, and 18 months. Secondary outcomes included the Cornell Scale for Depression in Dementia (CSDD), cognition, activities of daily living, resource use, and caregiver’s depression severity, and healthcare use.	Collaborative care group showed significant improvement in behavioral NPI scores and caregiver stress. There was no impact on depression scales, cognitive of functional status. Augmented usual care showed fewer cumulative physician and nurse visits. Rates of nursing home placement did not differ between groups.	23
Czaja et al., 2013	African American caregivers ≥ 21 yo of community-dwelling PWD located in Miami, FL * Non-English (Spanish) and multicultural populations included	Virtual (e.g., videophone-based)	*n* = 110 (total) *n* = 38 (intervention) *n* = 36 (attention control) *n* = 36 (control) * *n* = 56 (Hispanic) * *n* = 54 (Black/African American) * *n* not reported for Spanish speaking participants	Caregivers were randomized to three groups: 1. Intervention: certified interventionists taught problem solving strategies to deal with behaviors, stress management, healthy behaviors, and communication conducted through in-home visits. 2. Attention control: In-home and videophone visits focusing on nutrition 3. Information only control: Education material about dementia, caregiving, safety, and resources were mailed to participants followed by brief phone call.	Measurements of depression, caregiver burden, social support, and the caregivers’ perception of the caregiver’s experience were administered at baseline and 5 months post- randomization.	Caregivers in the intervention group compared to controls experienced decrease caregiver burden, increased appreciation of positive aspects of caregiving, and greater satisfaction with social support.	16

Key: PWD = persons with dementia. PMD = primary medical doctor. NPI = Neuropsychiatric Inventory. CSDD = Cornell Scale for Depression in Dementia. * = indicated special population.

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
