# Peer review of "Systematic Review of Dementia Support Programs with Multicultural and Multilingual Populations"

_geriatrics, 2021, doi:10.3390/geriatrics7010008_

Round 1
Reviewer 1 Report
I found this an interesting article, but please review your language as indicated in the comments below.
Abstract, line 22 – please do NOT use ‘suffering from dementia’. The language we use is important as it can reinforce negative stereotypes or promote a more positive view. ‘living with dementia’ would be preferable.
Lines 29-30 – as above, the language used is not acceptable, and it also too general as it implies everyone with dementia will have that experience. ‘Those living with dementia can experience negative outcomes including…’ would be better. Even if that’s the language used in any references, unless you are directly quoting (which you’re not) you should use appropriate language in your article.
Line 49 – you use ‘patients with dementia’ bit are they actually ‘patients’ (i.e. in hospital or similar), or ‘people’? You’ve uses ‘persons with dementia’ elsewhere, so unless there is a specific reason for using ‘patients’ I think you should not use it. Again, it’s thinking about the language used and the impact this can have on how people with dementia are viewed. (similar point for the Possin et al article in Table 1, as in the first column you refer to ‘community-dwelling persons with dementia’ but in the last column to say ‘patients with dementia’)
Line 98 – why ‘patient’s home’? If they’re at home, are they actually a ‘patient’?
Line 115 – this is my final time querying the use of ‘patients’, but please check the whole article to see whether the use of ‘patients’ is justified.
Line 144 – please remove the word ‘suffering’. As with ‘patients’, I won’t highlight any further instances but will assume that you check the whole article.
Lines 152-154 – I would be tempted to summarise this information in a table or graph, but if you leave it as text I think it would be better to list them in order of score, i.e. four “very good”, two “good”, none “fair”, one “poor” rather than starting by saying none were “fair”. It added an unnecessary bit of initial confusion on reading.
Lines 223-225 – ignoring the language used, I was hoping for more from the conclusion. Following on from the discussion, it would be nice to have a very brief summary to highlight if any aspects of the interventions seemed to work better/worse than others. If there’s nothing you can pull out like this, then a sentence to indicate the lack of consistency between the interventions would be useful. E.g., the programs help, but the small number of studies meant it was not possible to identify specific elements that were more/less beneficial… (not those words obviously, but that’s the general idea).
Minor points
Line 44 – it seems odd to use the present tense by saying ‘we aim to conduct’ when you’re reporting on something that you have done. ‘we aimed to conduct’ or ‘we conducted’ would seem more appropriate
Line 70 – I think that ‘were’ should be ‘was’
Lines 72-73 – you say that the tool was used to evaluate both randomized and non-randomized control trials, but you’ve previously said (line 56) that you looked at randomized. Where does the non-randomized come in? If you’re talking more generally that this is how the tool ‘is’ used, rather than specifically for this study, this needs to be made clear.
Figure 1 – The first filter box looks like it is too small as it cuts off partway through the second bullet point. Please check all of the boxes to ensure the figure is complete.
Some of the spacing looks odd throughout the article (e.g. large gap between sentences in line 69, but almost no gap in line 74) and in Table 1, but this may be the formatting rather than anything you can change. I don’t see the need to justify text on both the left and right in Table 1, especially as it makes it difficult to read.
Line 109 – If you’ve previously said ‘identified as Hispanic/Latino’ and ‘identifying as African Americans’ why is it ‘identifying Black/African American’? Should there be an ‘as’ included?
Line 157 – you’ve previously said that PWD is persons with dementia but here you say it is patients with dementia. Hopefully your review of the use of ‘patients’ will resolve this and ensure consistency.
Lines 173-174 – I’m not sure about the use of ‘In addition to’ and ‘additional’ in the same sentence. Is there a way to change one to be less repetitive?
Reviewer 2 Report
The manuscript aimed to evaluate the outcomes of dementia care programs that included multicultural and non-English speaking populations. The idea is interesting with few elements of novelty but the article is well-written. Nevertheless, the discussion and the reference section need to be improved. Please, discuss the role of comorbidities in older adults. Please cite and discuss the following PMID:
- 34775786
- 33030109
- 32213845
- 34882939
- 34440883
- 24931304
- 28222520
- 31927710
- 19366776
- 30149446
- 29409842
Round 2
Reviewer 2 Report
The discussion and the reference section need to be improved. The manuscript is not improved from the previous revision.